# VISUAL HIDE AND SEEK

## ABSTRACT

We train embodied agents to play Visual Hide and Seek where a prey must navigate in a simulated environment in order to avoid capture from a predator. We place a variety of obstacles in the environment for the prey to hide behind, and we only give the agents partial observations of their environment using an egocentric perspective. Although we train the model to play this game from scratch, experiments and visualizations suggest that the agent learns to predict its own visibility in the environment. Furthermore, we quantitatively analyze how agent weaknesses, such as slower speed, effect the learned policy. Our results suggest that, although agent weaknesses make the learning problem more challenging, they also cause more useful features to be learned.

## 1 INTRODUCTION

We introduce the task of *Visual Hide and Seek* where a neural network must learn to steer an embodied agent around its environment in order to avoid capture from a predator. Our hypothesis, which our experiments suggest, is that learning to play this game will cause useful representations of multi-agent dynamics emerge. We train the agent to navigate through its environment to maximize its survival time, which the model successfully learn to do. However, since we train the model from scratch to directly map pixels to action, the network can learn a representation of the environment and its dynamics. Figure 1 illustrates the problem setup.

We designed this game to mimic the typical dynamics between predator and prey. For example, we place a variety of obstacles inside the environment, which create occlusions that the agent can leverage to hide behind. We also only give the agents access to the first-person perspective of their three-dimensional environment. Consequently, this task is a substantial challenge for reinforcement learning because the state is both visual (pixel input) and partially observable (due to occlusions). To carefully study the emerged representations and agent dynamics, we probe different environmental factors as well as the agents' abilities, and our experiments quantitatively suggest that the agent automatically learns to recognize the perspective of the predator and its own visibility, which enables robust hiding behavior.

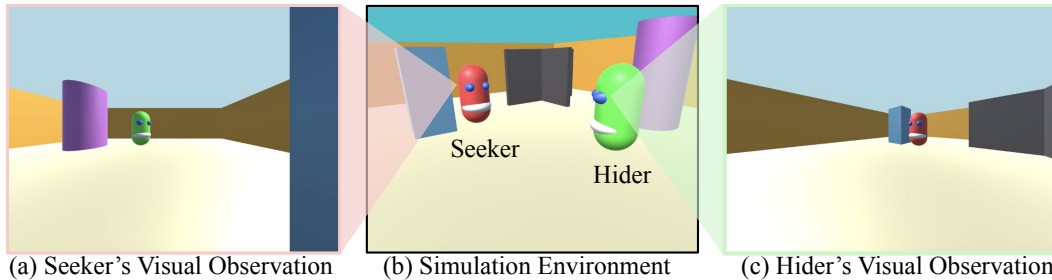

(a) Seeker's Visual Observation  (b) Simulation Environment  (c) Hider's Visual Observation

Figure 1: **Visual Hide-and-Seek:** We train models to play a game of visual hide and seek and analyze the dynamics that automatically emerge. We show that, although agent weaknesses make the learning problem more challenging, they collaterally encourage the learning of rich representations for the scene dynamics.

However, what intrinsic structure in the game, if any, caused this strategy to emerge? We quantitatively compare a spectrum of hide and seek games where we perturb the abilities of agents and the environmental complexity. Our experiments show that, although agent weaknesses, such as slower speed, make the learning problem more challenging, they also cause the model to learn more useful representations of its environmental dynamics. We show there is a "sweet spot" where the weakness is strong enough to cause useful strategies to emerge without derailing the learning process.

This paper makes three principal contributions to embodied agents. Firstly, we introduce the problem of visual hide-and-seek where an agent receives a partial observation of its visual environment and must navigate to avoid capture. Secondly, we empirically demonstrate that this task causes representations of other agents in the scene to emerge. Thirdly, we analyze the underlying reasons why these representations emerge, and show they are due to imperfections in the agent's abilities. The rest of this paper analyzes these contributions in detail. We plan to release all software, data, environments, and models publicly to promote further progress on this problem.

## 2 Related Work

Our work builds on research for embodied agents that learn to navigate and manipulate environments. Embodied agents with extensive training experience are increasingly able to solve a large number of problems across manipulation, navigation, and game-playing tasks (Mnih et al., 2015; Gu et al., 2017; Zhu et al., 2017b; Silver et al., 2017; Kahn et al., 2018; Kalashnikov et al., 2018; Mirowski et al., 2018). Extensive work has demonstrated that, after learning with indirect supervision from a reward function, rich representations for their task automatically emerge (Bansal et al., 2017; Lowe et al., 2017; Liu et al., 2019; Jaderberg et al., 2019). Several recent works have created 3D embodiment simulation environment (Kolve et al., 2017; Brodeur et al., 2017; Savva et al., 2017; Das et al., 2018; Xia et al., 2018; Savva et al., 2019) for navigation and visual question answering tasks. To train these models, visual navigation is often framed as a reinforcement learning problem (Chen et al., 2015; Giusti et al., 2015; Oh et al., 2016; Abel et al., 2016; Bhatti et al., 2016; Daftry et al., 2016; Mirowski et al., 2016; Brahmbhatt & Hays, 2017; Zhang et al., 2017a; Gupta et al., 2017a; Zhu et al., 2017a; Gupta et al., 2017b; Kahn et al., 2018). Moreover, by incorporating multiple embodied agents into the environment, past work has explored how to learn diverse strategies and behaviors in multi-agent visual navigation tasks (Jaderberg et al., 2019; Jain et al., 2019). For a full review of multi-agent reinforcement learning, please see (Panait & Luke, 2005; Bu et al., 2008; Tuyls & Weiss, 2012; Shoham et al., 2007; Hernandez-Leal et al., 2018).

Our paper contributes to a rapidly growing area to learn representations from indirect supervision. Early work has studied how features automatically emerge in convolutional networks for image recognition (Zhou et al., 2014; Zeiler & Fergus, 2014). Since direct supervision is often expensive to collect, there has been substantial work in learning emergent representations across vision (Doersch et al., 2015; Vondrick et al., 2018), language (Kottur et al., 2017; Radford et al., 2017), sound (Owens et al., 2016; Aytar et al., 2016), and interaction (Aytar et al., 2018; Burda et al., 2018). We also study how representations emerge. However we investigate the emergent dynamics in the two-player game of visual hide and seek. We characterize why representations emerge, which we believe can refine the field's understanding of self-supervised learning dynamics.

This paper is concurrent to (Baker et al., 2019), and we urge readers to watch their impressive results on learning to play hide and seek games. However, there are a few key differences between the two papers that we wish to highlight. Firstly, in contrast to (Baker et al., 2019), we focus on hide and seek in partially observable environments during both training and testing. Our environment is three-dimensional, and agents only receive an egocentric two-dimensional visual input, which creates situations abundant with occlusions. Secondly, the input to our model is a visual scene, and not the state of a game engine. The learning problem is consequently very challenging because the model must learn perceptual representations in addition to its policy. Our experiments suggest this happens, but the richness of the visual representation depends on the impediments to the model. Finally, we focus our investigation on analyzing the underlying reasons *why* different behaviors emerge, which we believe will refine the field's insight into self-supervised learning approaches.

# 3 HIDE AND SEEK

We first present our environment and learning problem, then describe our interventions to understand the cause of different emergent behaviors.

## 3.1 ENVIRONMENT AND LEARNING

We created a 3D simulation for hide-and-seek using the Unity game engine, which we use throughout this paper. There are two agents in this game: the hider and the seeker. Each agent receives only a first-person visual observation of the environment with 120-degree field of view, and navigates around the environment by selecting actions from a discrete set (move forward, move backward, turn left, turn right, stand still). The environment is a square of 14 unit x 14 unit, and any real value position inside the square without being occupied any obstacle is a valid game state. The speed of the hider is two units per second while the speed of the seeker is one and a half units per second. Our simulation runs in 50 frame per second. The agents can turn 3.3 degree per frame. Each agent has a diameter of one unit. In contrast to other multiplayer games (Sukhbaatar et al., 2016; Moravčík et al., 2017; Lample & Chaplot, 2017; MacAlpine & Stone, 2017; Lowe et al., 2017; Foerster et al., 2018), the egocentric perspective of our problem setup makes this task very challenging. We place obstacles throughout the environment to create opportunities for occlusion.

*Seeker Policy:* We use a simple deterministic policy for the seeker, which is as follows. If the seeker can see the hider, move towards it. If the seeker cannot see the hider, move towards the hider's last known position. If the seeker still cannot find the hider, then it will explore the environment by waypoints in a round-robin fashion. The game episode concludes when the seeker successfully "catches" the hider. We define catching the hider as a collision between the two agents.

*Hider Policy:* Given the environment and the seeker algorithm, we train a policy for the agent to hide. Given a first-person observation of the agent's environment, we estimate the action to execute with a convolutional neural network. We train the model to maximize its reward function using PPO (Schulman et al., 2017). Each time step, we provide a small reward ($+0.001$) for each living step, and a large negative reward ($-1$) once captured. We call this model **basic**.

*Starting Positions:* The starting positions of the hider and the seeker are randomly sampled on a continuous grid for each episode. We also specify their initial orientations so that they always start by looking towards each other (even if there is an obstacle between them).

Our intention is that, by learning to hide, the model will learn features for the recognition of objects in its visual field, such as obstacles and other agents. To analyze this, we fit a linear regression from the learned features to classify different objects and game states. We encode object and game states as discrete variables, and use the classification accuracy as a measure of how well the features are encoding different states. Our experiments will use this approach to probe the learned features.

## 3.2 SYSTEMATIC INTERVENTIONS

We systematically intervene on the learning process to understand the mechanisms behind emergent features. We chose these interventions because they either give the agent an advantage or weakness. We summarize these variations in Table 1. By training the models from scratch with different interventions and analyzing the learned representation, we can analyze how environmental features cause different strategies to be learned.

*Speed:* The first intervention we will make is manipulating the relative speed of the hider and seeker. We explore two variations. In the first variant, we slow down the hider. Likewise, in the second variant, we speed up the hider. We call these agents **slowerhider** and **fasterhider** respectively. The conditions are otherwise the same as the basic model.

*Self-Visibility:* Awareness of your own visibility is a crucial feature for successfully surviving during hide and seek (Russell et al., 2012; Reinhold et al., 2019). We explicitly incorporate this into the reward function as an auxiliary dense reward signal. Along with the sparse reward above, this agent receives an additional reward of $0.001$ if the seeker does not see it at the current time step. Likewise, the agent receives a punitive reward of $-0.001$ if it is currently visible by the seeker. We refer to this

agent as **visibilityreward**. Furthermore, we also use a variant where the hider has a faster speed, which we call **visibilityreward+faster**.

*Environmental Complexity:* We also intervene on the complexity of the environment, which lets us analyze the impact of the environment on the emergent features. We use two variations. Firstly, we use a stochastic policy for the seeker agent (instead of deterministic). Specifically, the stochastic seeker randomly visits locations in the map until the hider is within its field of view, at which point it immediately goes towards it. We name this variation **stochasticseeker**. Secondly, we use a stochastic map where we randomly select a number of objects and also randomly position throughout the environment. Consequently, the difficulty of the maps will change between easy (many occlusions) to difficult (few to none occlusions). We name this variation **stochasticmaps**.

| Hider Name | Speed | Seeker Policy | Maps | Visibility Reward |
|---|---|---|---|---|
| basic | 2 | Deterministic | Deterministic | None |
| fasterhider | A | Deterministic | Deterministic | None |
| slowerhider | 1 | Deterministic | Deterministic | None |
| stochasticseeker | 2 | Stochastic | Deterministic | None |
| stochasticmaps + stochasticseeker | 2 | Stochastic | Stochastic | None |
| visibilityreward | 2 | Deterministic | Deterministic | Yes |
| visibilityreward + faster | A | Deterministic | Deterministic | Yes |

Table 1: **Interventions:** We perturb the learning process in order to understand the causes of different strategies during visual hide and seek. When the speed is "A", the agent has allowed to accelerate to reach higher speeds at a rate of two units per time period squared.

### 3.3 IMPLEMENTATION DETAILS

We implement our simulation using the Unity game engine along with ML-Agents (Juliani et al., 2018) and PyTorch (Paszke et al., 2017). We plan to publicly release our framework. Given an image as input, we use a four-layer convolutional network as a backbone, which is then fed to a two-layer fully connected layers for the actor and critic models. We use LeakyReLU (Maas et al., 2013) as activation function throughout the network. The input pixel values are normalized to a range between 0 and 1. We train the network using Proximal Policy Optimization, which has been successful across reinforcement learning tasks (Schulman et al., 2017). We optimize the objective using Adam optimizer (Kingma & Ba, 2014) for $8 * 10^6$ steps with a learning rate $3.0e - 4$ and maximum buffer size of 1,000. We then rolled out the policies for 100 episodes using the same random seed across all the variants. Each episode had 1,000 number of steps at maximum. Following (Mnih et al., 2015), we use a 6-step repeated action, which helps the agent explore the environment. Please see the appendix for full details.

## 4 EXPERIMENTS

The goal of our experiments is to analyze the learned capabilities of the polices, and characterize why strategies emerge in hide and seek games.

Overall, our experiments show that the hiding agent is able to learn to avoid capture. We show that each policy learns a different strategy depending on its environmental abilities. During the process of learning to hide, our results suggest that the agent automatically learns to recognize whether itself is visible or not. We also found that with judicious weakness, the model learns to overcome its disadvantage by learning rich features of the environment. The rest of this section investigate these learning dynamics.

### 4.1 DOES THE AGENT LEARN TO SOLVE ITS TRAINING TASK?

We first directly evaluate the models on their training task, which is to avoid capture from the seeker. We quantify performance using two metrics. Firstly, we use the average number of living steps across all testing episodes. Secondly, we use the success rate, which is the percentage of games that the hider avoids capture throughout the entire episode. Table 2 reports performance, and shows that

| Environment | Average of Living Steps (trained / random) | Success Rate (trained / random) |
|---|---|---|
| basic | $513 \pm 10$ / $43 \pm 3$ | $46.02\% \pm 2.60\%$ / 0 |
| fasterhider | $522 \pm 3$ / $43 \pm 5$ | $33.27\% \pm 2.00\%$ / 0 |
| visibilityreward + faster | $422 \pm 24$ / $41 \pm 4$ | $33.29\% \pm 1.33\%$ / 0 |
| stochasticseeker | $271 \pm 5$ / $43 \pm 2$ | $23.75\% \pm 1.52\%$ / 0 |
| stochasticmaps + stochasticseeker | $281 \pm 12$ / $55 \pm 7$ | $17.45\% \pm 1.59\%$ / 0 |
| visibilityreward | $248 \pm 13$ / $46 \pm 2$ | $19.93\% \pm 3.55\%$ / 0 |
| slowerhider | $72 \pm 5$ / $44 \pm 3$ | $1.08\% \pm 0.07\%$ / 0 |

Table 2: **Hider Performance:** We show the success rate and average number of living steps of different Hider agents and environments. The maximum number of steps for each episode is 1,000.

all agents solve their task better than random chance, suggesting the models successfully learned to play this game. As one might expect, the stronger the agent, the better the agent performed: the faster agent frequently outperforms the slower agent.

## 4.2 WHAT VISUAL CONCEPTS ARE LEARNED IN REPRESENTATIONS?

In this experiment, we want to investigate which visual concepts are encoded in the latent representation. For example, learning to recognize the predator and whether the agent is itself visible to the predator are important prerequisites for hide and seek strategies. Consequently, we design two corresponding tasks to study the encoded visual concepts from the trained policies. The first task analyzes whether the learned features are predictive of the seeker (**Seeker Recognition**). The second analyzes whether if the hider is able to infer the visibility of itself to the seeker (**Awareness of Self-Visibility**). These two tasks can be categorized as two binary classification downstream tasks. To do this, we extract the mid-level features from the learned policy network in different environments, and train a binary logistic regression model on these two proposed tasks.

| Environment | Seeker Recognition ( /random init policy) | Awareness of Self-Visibility ( /random init polcy) |
|---|---|---|
| basic | $91.03 \pm 0.21$ / $79.25 \pm 2.42$ | $75.84 \pm 0.66$ / $63.92 \pm 0.50$ |
| fasterhider | $90.01 \pm 0.34$ / $74.84 \pm 1.91$ | $77.88 \pm 0.05$ / $60.30 \pm 0.13$ |
| visibilityreward + fasterhider | $79.19 \pm 1.11$ / $74.18 \pm 1.95$ | $65.05 \pm 1.88$ / $60.84 \pm 0.84$ |
| stochasticseeker | $96.17 \pm 0.25$ / $77.54 \pm 1.29$ | $94.00 \pm 0.58$ / $65.05 \pm 0.13$ |
| stochasticmaps + stochasticseeker | $95.55 \pm 0.38$ / $80.75 \pm 1.67$ | $94.25 \pm 0.75$ / $70.80 \pm 0.38$ |
| visibilityreward | $96.21 \pm 0.46$ / $77.65 \pm 0.15$ | $95.30 \pm 0.38$ / $63.88 \pm 2.30$ |
| slowerhider | $83.96 \pm 0.46$ / $77.79 \pm 0.54$ | $81.71 \pm 0.63$ / $68.59 \pm 0.34$ |

Table 3: **Downstream Visual Perception Tasks:** the table shows the classification accuracies on two important visual perception tasks using mid-level features from all the trained policies. Regardless of the advantages of the hider agent and high performance in the hiding task, the "visibilityreward + faster" policy has the worst performance on both of the two tasks.

The extracted feature is the activation from the last convolutional layer in the policy network with a dimension of 512. The labels are from ground-truth game states. We emphasize that we do not use any of these labels during learning. We only rely on them for studying the representations after the training is completed. Throughout the paper, we use "S" and "¬S" to represent the game state of whether the seeker is visible to the hider, and "H" and "¬H" to represent the game state of whether the hider is visible to the seeker. Therefore, the inputs of these two tasks are the visual observations of the hider agent, and the labels are "H"/"¬H" and "S" /"¬S" respectively. We then use the classification accuracy on the test images as a measurement on how well the latent representations encode corresponding visual concepts. We rolled out each learned policy for 6,000 steps for feature extraction and use a 0.8 train and test split.

Table 3 shows the results for both tasks. The faster agent, especially "visibilityreward + faster", performs worst on these two perception tasks, even though it has the most advantages among all the

variants. Since the faster agents performed better at the survival task (Table 2), this suggests that the performance in the training task does not directly cause the performance in other downstream tasks. To further demonstrate this, we visualize the embedding mid-level features of "visibilityreward" and "visibilityreward+faster" policy with respect to the labels of the above two tasks using t-SNE (Maaten & Hinton, 2008) in Figure 2. The features for "visibilityreward" policy is more separable than the features for "visibilityreward + faster". This qualitative evidence supports that the "visibilityreward + faster" policy is worse at downstream visual perception tasks even though the hider in this case can move a lot faster.

In the second task, we analyze whether the hider is able to infer its own visibility to the seeker. The features from the stochastic variants and the "visibilityreward" are the most predictive of the hiders own visibility. This could be because the hider agent is encouraged to pay more frequent attention to the seeker agent in order to get higher rewards.

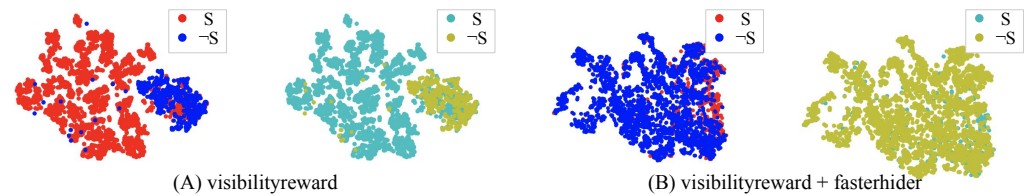

|  (A) visibilityreward  |  (B) visibilityreward + fasterhider  |

Figure 2: **t-SNE Embedding** of mid-level features colorized by the labels of the two visual perception tasks from Section 4.2.

### 4.3   WHAT CAUSES USEFUL FEATURES TO EMERGE?

To analyze the relationship between training task difficulty and emerging skills, Figure 3 compares the classification accuracy of using the features to predict self-visibility versus the agent's survival time. Each dot in the plot represents one variant of the environments specified in Table 1. The corresponding value of each dot is drawn from Table 2 and Table 3. If the agent is not able to avoid capture at all, then the features are clearly poor. However, if the agent is able to completely evade capture, then the features are also poor! Instead, there is a concave relation that suggests agent weaknesses are actually advantageous for learning rich representations. We believe this is the case because by giving the model a disadvantage, the learning process will compensate for the weakness, which it does by learning stronger features.

### 4.4   HOW DOES THE TRAINED MODEL HIDE?

Each agent is trained with varying advantages and disadvantages. We are interested in understanding how these variations affect the behavior of learned policy and how exactly the trained model learns to hide. We quantitatively analyze the *dynamics* of the learned policies by probing the internal game engine state. Note that we only use the game engine states for analysis, not learning.

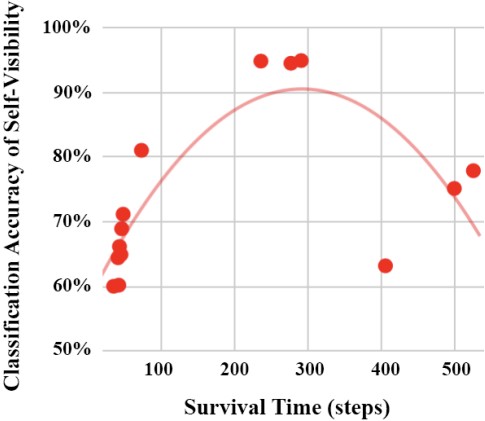

Figure 3: **Quality of Representation vs. Survival Time**: We compare the performance of models on their survival time versus how well the internal representation is predictive of downstream recognition tasks. Each red dot represents one policy trained with different advantages or disadvantages. The curve is the parabolic best fit. Interestingly, improved task performance (survival time) does not always create stronger representations. Instead, the best representations arise from the models with intermediate disadvantages. When the model has a weakness, the model learned to overcome it by instead learning better features.

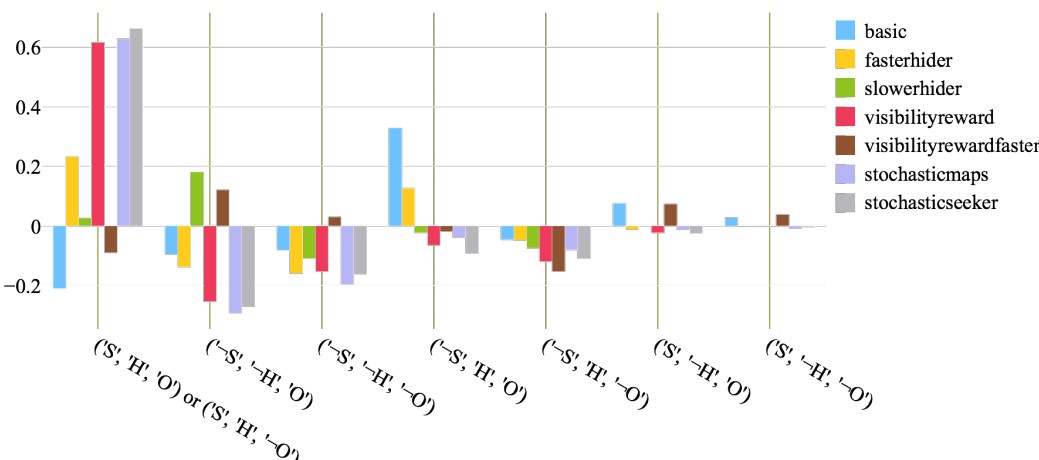

Figure 4: **Frequencies of Visual States.** We show the relative frequency over chance that each visual perception states are visited. "S" stands for whether the Seeker is visible to the Hider, "H" stands for whether the Hider is visible to the Seeker, and "O" means whether there is any obstacle visible to the Hider. "¬" denotes the opposite condition. This plot demonstrates a quantitative way to measure different behaviors using visiting frequencies among important states. For example, we can tell that "basic" agent learns to run with its back facing the seeker as shown in the high blue bar for state ('¬S', 'H', 'O'). The "basic" hider learns to always run away because its similar speed with the seeker. The "slowerhider" favors to stay at ('¬S', '¬H', 'O') where the hider and the seeker cannot see each other and there is at least one obstacle in front itself. This suggests the slower hider is learning to leverage the environment to compensate for its speed disadvantage. "visibilityrewardfaster" often stays either at the state where they cannot see each other as suggested by the high brown bars of state ('¬S', '¬H', 'O') and ('¬S', '¬H', '¬O'), or stays at the state where the it can see the seeker but the seeker cannot see itself as indicated in the high brown bars at ('S', '¬H', 'O') and ('S', '¬H', '¬O'). This suggests that the "visibilityrewardfaster" can use its speed to hide completely and check the status of the seeker while keeping itself safe.

*Frequency of Visual States:* To analyze states for each policy, we track three visual states in the hide and seek game. 1) **"S" / "¬S"**: whether the hider can see the seeker, 2) **"O" / "¬O"**: whether the hider can see any obstacle and 3) **"H" / "¬H"**: whether the hider is visible to the seeker. We rolled out each learned policy for 50,000 steps, and counted how often they enter these game states. Then we plot their relative frequencies by subtracting the frequencies of corresponding random initialized policies from the absolute visiting frequencies.

Figure 4 compares the frequency of states for each model. We observe a few key differences among the agents. When the speeds and abilities of the hider and seeker are the same (basic), the hider learns a policy to first turn away from the seeker then run away, as evidenced by the high blue bar for the state (¬S, H, O). However, when the hiding agent is slower than the seeker, the hider frequently enters a state where the two agents cannot see each other, but an obstacle is visible, as evidenced by the high green bar for the state (¬S, ¬H, O). In other words, when the hiding agent has a disadvantage, the model cannot rely on its intrinsic capabilities, and will instead learn to leverage the environment to perform its task. In contrast, when the hiding agent has several advantages such as "visibilityrewardfaster", the policy also learns to make use of them during the training process. As shown in the high brown bars at state ('¬S', '¬H', 'O') and ('¬S', '¬H', '¬O'), as well as ('S', '¬H', 'O') and ('S', '¬H', '¬O'), this hider learns to use its speed and auxiliary visibility reward to hide completely from the sight of the seeker and monitor the status of the seeker while keeping itself safe at the same time.

*Probability of State Transitions:* While the state frequency shows the agents' most common states, it does not reflect the *dynamics* of the decision making process. We use the state transitions to further analyze the agent behaviors. For example, a transition from "H" to "¬H" indicates a successful hiding attempt. Following the previous state definitions, there are eight possible combinations out

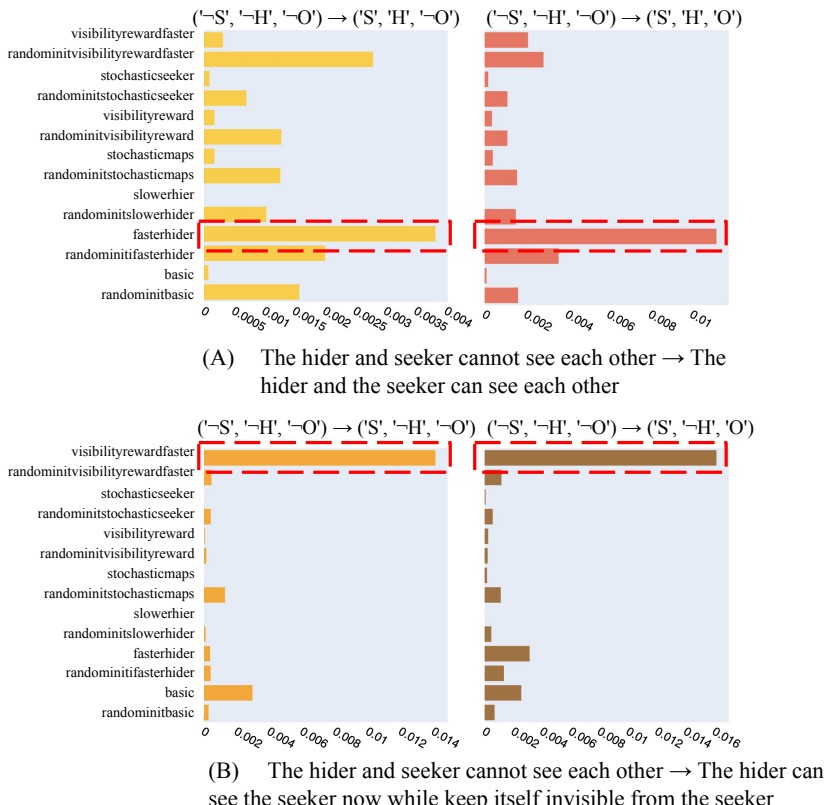

(A)  The hider and seeker cannot see each other → The hider and the seeker can see each other

(B)  The hider and seeker cannot see each other → The hider can see the seeker now while keep itself invisible from the seeker

Figure 5: **Transition Dynamics:** We show the probability of transitions between states across all the variants. All of the four plots share the same y-axis in the table above. Each bar indicates the probability of transiting from one state to the other. The state transitions are labeled on top of each graph, and an explanation of the transition on the bottom of each graph. "randominit" prefix represents the corresponding random initialized policy for each variant.

of these three states and $(8 \times 8 - 8) = 56$ transitions in total excluding identity state transitions. We summarize representative cases, and will release the full results for all combinations along with our code bases.

Figure 5 quantifies different exploration strategies for each model. For example, the top tow (A) suggests that faster agents tend to monitor the status of the seeker while exposing itself. This likely happens because the faster agents are fast enough that they can immediately react when the seeker becomes too close. Specially, the two red bars in (A) denotes that "fasterhider" policy has the highest probability to transition from the state where the hider and the seek *cannot* see each other to the state where the hider and the seeker *can* see each other. When the status of the seeker becomes clear, the hider in "fasterhider" also exposes itself to the seeker's view.

However, when the faster agent is explicitly penalized for exposing itself ("visibilityreward + fasterhider"), the agent tends to monitor the status of the seeker while keeping itself hidden. This is suggested by the two red bars from Figure 5 (B) where "visibilityreward + fasterhider" policy has the highest probability to transit from the state where the hider and the seeker *cannot* see each other to the state where the hider can see the seeker, yet the seeker *cannot* see the hider. In these cases, the agent has learned to watch the predator, but remain outside of its field of view.

*Distance between Agents with respect to Time:* We also measure the learned behavior by quantifying distances between the two agents with respect to time. In contrast to previous analysis which focused on short-term dynamics, this lets us quantify long-term dynamics.

Figure 6 plots the distance between agents versus time, alongside representative qualitative cases. When the two agents have the same capabilities (Figure 6a), the agents gradually get closer to each

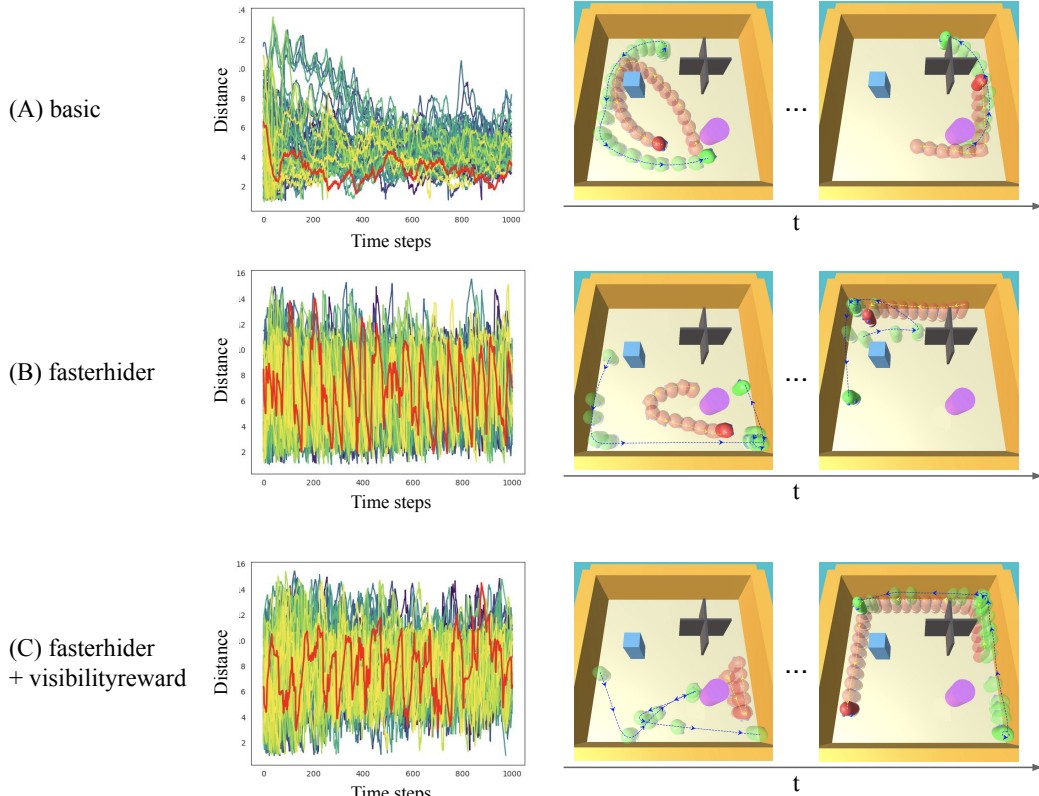

Figure 6: **Distance Versus Time:** We use distance between agents with respect to time steps to quantify long-term dynamics of the game. Each rollout trajectory is represented by one constant color over time. We show multiple trajectories to reflect the general pattern for different policies. Hider agents from different policies exhibit diverse pattern in this plot. We further append qualitative demonstrations to explain what dynamics each pattern correspond to on the right side of the plots. The corresponding trajectory is plotted in red line.

other. However, when one of the hiding agent is faster, the distance between the hider and seeker will significantly oscillate (Figure 6b,c). This is consistent with a reactive strategy where faster agents are able to stand still until they are threatened, at which point they simply run away.

Next to the plots, we also show visual demonstrations that depicts the correlated strategies learned by different hider agents. The corresponding trajectory is colored in red. One key difference is between the faster hiders with and without a visibility dense reward. The "fasterhider" moves towards the seeker while facing it at the same time, then it moves away when they get too close. On the other hand, the "fasterhider + visibilityreward" agent moves towards the seeker by monitoring the seeker from behind.

## 5 DISCUSSION AND FUTURE WORK

Our experiments suggest there are many diverse strategies for learning to hide from a predator. Moreover, during the learning process, the agents learn a visual representation for their task, such as recognizing their own visibility. However, our experiments show that this emergent representation requires a judicious disadvantage to the agent. If the weakness is too severe, then the learning is derailed. If there is no weakness or even an advantage, then the learning just discovers a reactive policy without requiring strong representations. However, with moderate disadvantages, the model learns to sufficiently overcome its weakness, which it does by learning strong features.

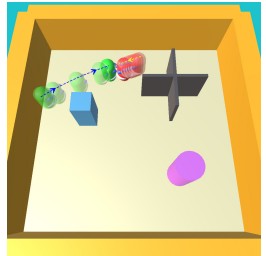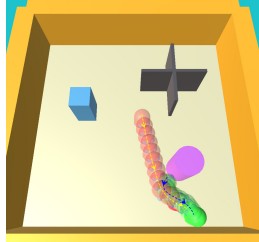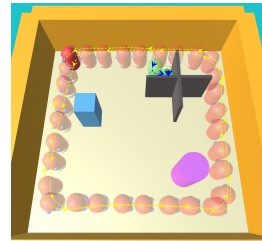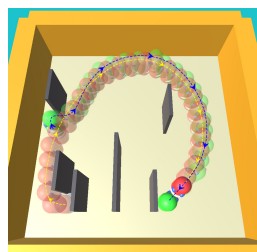

(a) **fasterhider** the Hider moved towards the Seeker quickly, then got caught

(b) **slowerhider** the Hider tended to move backward, then died due to no reaction time

(c) **slowerhider** the Hider found a design flaw in the game, then stay there to live forever

(d) **stochasticmaps** the Hider was caught because it did not notice there was an obstacle behind it

Figure 7: **Representative Failures:** We show a few cases where the hider is unable to avoid capture. Many of these failures are due to lack of memory in the model, suggesting improved memory representations will help this task.

We believe visual hide and seek, especially from visual scenes, is a promising self-supervised task for learning multi-agent representations. Figure 8 shows a few examples where the hider fails to escape from the predator. Many of these failures are due to the lack of memory in the agent, both for the memory of the map and memory of previous predator locations, which invites further research on computational memory models (Milford et al., 2004; Gupta et al., 2017a; Zhang et al., 2017b; Parisotto & Salakhutdinov, 2017; Khan et al., 2017; Wayne et al., 2018). Overall, these results suggests that improving the agent's ability to learn to hide while simultaneously increasing the severity of the disadvantage will cause increasingly rich strategies to emerge. With more complicated environment set up such as more complex room layout, diverse object categories and allowance for agents and objects interaction, we believe more visual representations and different level of game dynamics can be studied in future research.

Furthermore, we hope that our work can also encourage future works on specialized algorithms for visual egocentric multi-agent learning. Group of agents with different goals and capabilities can cause the environment become highly non-stationary. Better policy learning algorithms and feature extraction models are likely to improve the learned strategies and representations. We hope that our work can step closer to study these future research topics and can open up interesting ideas and directions.

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

# A APPENDIX

## A.1 NETWORK ARCHITECTURE

| Layer | Kernel Size | Num Outputs | Stride | Padding | Dilation | Activation |
|---|---|---|---|---|---|---|
| Conv1 | 8 | 32 | 4 | 0 | 1 | LeakyReLU |
| Conv2 | 4 | 64 | 2 | 0 | 1 | LeakyReLU |
| Conv3 | 3 | 64 | 1 | 0 | 1 | LeakyReLU |
| FC1 | N/A | 512 | N/A | N/A | N/A | LeakyReLU |
| Actor-FC2 | N/A | 512 | N/A | N/A | N/A | LeakyReLU |
| Actor-FC3 | N/A | 5 | N/A | N/A | N/A | LeakyReLU |
| Critic-FC2 | N/A | 512 | N/A | N/A | N/A | LeakyReLU |
| Critic-FC3 | N/A | 1 | N/A | N/A | N/A | LeakyReLU |

Table 4: **Policy Network Architecture** We list all the parameters used in the policy network.

## A.2 MORE VISUALIZATIONS FOR HIDING BEHAVIORS

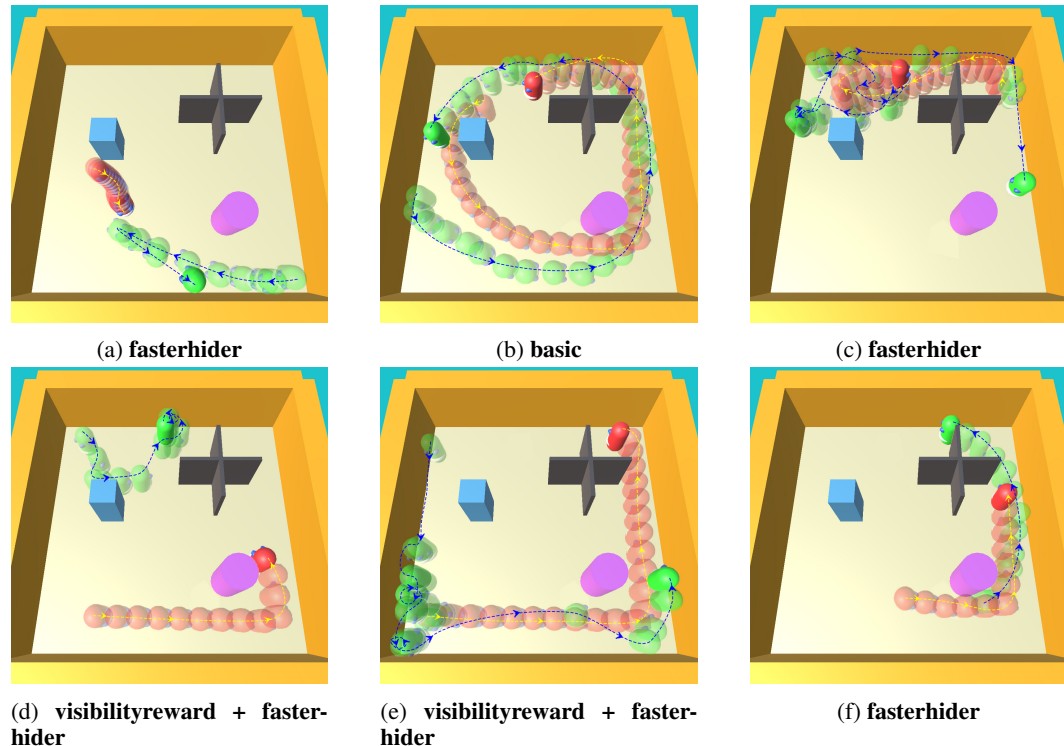

(a) **fasterhider**

(b) **basic**

(c) **fasterhider**

(d) **visibilityreward + faster-hider**

(e) **visibilityreward + faster-hider**

(f) **fasterhider**

Figure 8: **More Visual Demonstrations**

