# OpenReview forum: "Visual Hide and Seek"
_ICLR.cc/2020/Conference — Reject_

### Official Review · AnonReviewer1 · 2019-10-11
**Official Blind Review #1**

**Rating:** 3

**Review:**


Paper Summary: The paper studies what an embodied agent that is trained using RL learns in the context of the hide and seek game. There are two agents in a simple environment (one hider and one seeker). The strategy for the seeker is fixed, but the hider learns a policy based on ego-centric visual input. The paper attempts to analyze how the learned representation differs with different capabilities of the agents or environment structure.

Originality:

The study of the learned representations in the context of hide and seek is new. There is concurrent work on hide and seek (e.g., Baker et al., 2019), but unlike this paper, they have access to groundtruth location information of the agents and it is not based on visual input.

Quality:

- The conclusions of the paper are either counter-intuitive or are based on some hypotheses and guesses.

For example, it is strange that "visibilityreward" performs worst for "Awareness of Self-Visibility". It is exactly trained for that task.

There are conclusions about "temporal events" (e.g., the hider learns to first turn away from the seeker then run away) just based on a single frequency number (Figure 4). There are many other possibilities that can result in the same frequency. If the authors draw conclusions about temporal events, they should show how values change over time. Single frequency numbers cannot be used for such conclusions.

- To check the learned representation, the common practice is to use features on a very different task. However, this paper addresses only two tasks "Seeker Recognition" and "Awareness of Self-Visibility" which are very close to the original task. One of the models even receives explicit rewards for the latter case.

- No standard deviation is reported for the results. Given the random nature of RL algorithms, standard deviations should be reported.

Significance:

I expected a more rigorous analysis since this is an analysis paper and there is no new methodology in the paper. Conclusions that are just based on a guess about some qualitative result or based on two state changes as in Figure 5 are groundless. More thorough analysis based on quantitative results are required to justify the claims and provide generalizable conclusions.

Clarity:

The paper is mostly clear, but there is some missing information:

- In Figure 3, which dot corresponds to which model? It is claimed that "When the model has a weakness, the model learned to overcome it by instead learning better features". To justify that, we need to know what the dots are.

- What do "randominits" in Figure 5 correspond to?

- What is red, yellow, blue and green plots in Figure 6? What is the standard deviation?

**Experience Assessment:**

I have published in this field for several years.

**Review Assessment: Checking Correctness Of Derivations And Theory:**

N/A

**Review Assessment: Checking Correctness Of Experiments:**

I carefully checked the experiments.

**Review Assessment: Thoroughness In Paper Reading:**

I read the paper thoroughly.

---

> ### Author Response · Authors · 2019-11-10
> **Responses to Official Review #1**
>
> Thank you for the helpful review! We are glad you appreciate the novelty of egocentric hide and seek. We have updated our paper to reflect all the changes and please kindly find our responses for each point below:
>
> - “The conclusions of the paper are either counter-intuitive or are based on some hypotheses and guesses. For example, it is strange that "visibilityreward" performs worst for "Awareness of Self-Visibility". It is exactly trained for that task.”
>
> We respectfully disagree.
>
> One of the main goals of our paper is to establish quantitative matrices to measure RL agents’ behavior, and all of our results are supported by both quantitative and controlled experiments. Our paper empirically compares seven different models on five different quantitative evaluations to study the emergent representations and dynamics.
>
> While might not be intuitive, our results are still explainable. The agent with visibility reward does not get the chance to learn features of self-visibility because of the limited speed hence the model received samples with significantly less variation of its self-visibility, which makes learning to discriminate self-visibility difficult. Therefore, this result serves as a control experiment between the “basic”, “fasterhider”, and “visibilityreward+faster” variants in Section 4.1, 4.2, and 4.3.
>
> - “There are conclusions about "temporal events". There are many other possibilities that can result in the same frequency. If the authors draw conclusions about temporal events, they should show how values change over time. Single frequency numbers cannot be used for such conclusions.”
> We believe there is a misunderstanding, which we wish to clarify. Figure 4 plots how often the agent visits each state. Figure 5 plots how often the agent transitions between states, which quantifies the temporal events because it does show the state values change over time. For example, Figure 5 shows the “visibilityrewardfaster” agent frequently transitions between “the two agents cannot see each other” —> “the hider can see the seeker, but not the other way around.”  We updated the paper to make this clear. Thanks for pointing this out.
>
> - “To check the learned representation, the common practice is to use features on a very different task. However, this paper addresses only two tasks "Seeker Recognition" and "Awareness of Self-Visibility" which are very close to the original task. One of the models even receives explicit rewards for the latter case.”
> Our results show that the model learns useful features for subtasks that are useful for visual hide and seek. However, it is not reasonable for the model to learn features for tasks uncorrelated with the training task. We speculate that as the environment becomes more complex, more subtasks will emerge, but this is out-of-scope. Our main result is to study the emergent behaviors of visual hide and seek, which our main experiments do.
>
> - We update the results to include the standard deviation. Since the relative ranking of models did not change, our main conclusions still hold.
>
> - “I expected a more rigorous analysis since this is an analysis paper and there is no new methodology in the paper. Conclusions that are just based on a guess about some qualitative result or based on two state changes as in Figure 5 are groundless.”
> Our analysis is rigorous because they are supported by empirical experiments. Figure 5 quantitatively evaluates transitions between agent states.
>
> - “In Figure 3, which dot corresponds to which model?”
> The corresponding model can be found by looking at the value of the survival time (x-axis) and classification accuracy (y-axis) in Figure 3. The same values appear in Table 2 and Table 3. We plot the results in Table 2 and Table 3 from a different perspective and dimension in order to study the correlation between the performance on visual perception tasks and the performance on the game itself. We update our paper in Section 4.3 to clarify this point.
>
> - “What do "randominits" correspond to?”
> “randominit” stands for the corresponding random initialized policy for each variant. We update the caption in Figure 5 to clarify this point.
>
> - “What is red, yellow, blue and green plots in Figure 6? What is the standard deviation? ”
> In Figure6, we show the relationship of the distance (y-axis) between the hider and seeker versus time (x-axis). Each color represents one trajectory and we plot 100 trajectories using different colors. We show all of these trajectories together to show the general pattern. We show one single red line as a representative case and visualize its corresponding trajectory on the left as indicated in the caption. We update the caption of Figure 6 along with the corresponding texts to reflect the above explanations. Since all the 100 random rollouts are plotted in the figure, there is no standard deviation to report.
>
> Thank you very much for your constructive comments. Please kindly let us know if we address your concerns. Thank you!

---

> > ### Author Response · Authors · 2019-11-13
> > **Responses and Revisions**
> >
> > Dear Reviewer 1,
> >
> > Thank you again for your helpful reviews and comments! We have refined our paper based on them. Specifically, we clarify the facts in our texts and figure captions. We also updated the tables to include the standard deviations.
> >
> > As the discussion period is about to end, please do not hesitate to kindly let us know if there are any additional clarifications or explanations we can offer, as we would love to convince you of the values of our paper. We appreciate your suggestions and comments! Thank you!

---

### Official Review · AnonReviewer2 · 2019-10-22
**Official Blind Review #2**

**Rating:** 3

**Review:**

Visual Hide and Seek

In this paper, the authors propose a two-agent hide and seek environment, and uses reinforcement learning to train the hider. The authors interpret the learned representation by using the learnt features to do classification.

I tend to vote rejection for this paper, mostly due to my feeling that the empirical contribution is interesting, but not novel enough for this conference.
There is still a lot of potential improvement available for this project, which is summarized below.

Pros:
- The environment itself, once open-sourced, can be quite valuable to the community.
I think it is fair to say that the proposed environment in this project is better than the hide-and-seek environment from OpenAI in that it provides the visual input.
By the way, I believe OpenAI environment is also partially observable, where unseen agents’ information is masked out.
- It is an interesting finding on how meaningful features and performance correlate with each other.

Cons:
- Findings in the project are very practical and interesting, but they do not seem to provide valuable information for future research.
For example, to improve the quality of the paper, is it possible to utilize the features and performance correlation to improve unsupervised visual feature learning?
Is it possible to draw a connection to Psychology? Does this “Representation vs. Survival Time” also happens in real-life, or is it just some empirical things that happen in optimization?

- There are a lot more potential to improve the environments. It will greatly increase the impact of the paper by, for example, considering multi-agent training, self-play, unsupervised training.

In general, I think this project still have a lot of room for fine-tuning.

**Experience Assessment:**

I have published in this field for several years.

**Review Assessment: Checking Correctness Of Derivations And Theory:**

I carefully checked the derivations and theory.

**Review Assessment: Checking Correctness Of Experiments:**

I carefully checked the experiments.

**Review Assessment: Thoroughness In Paper Reading:**

I read the paper thoroughly.

---

> ### Author Response · Authors · 2019-11-10
> **Responses to Official Review #2**
>
> Thank you for your constructive points! We are glad you appreciate the proposed task (visual hide and seek) as well as the experimental results. To promote further research progress on this problem, we will publicly release all the software and code. We have updated our paper to reflect all the changes and please kindly find our responses for each point below:
>
> - Since this paper is the first to propose and empirically investigate the task of visual hide and seek, we believe the contribution is substantial and will receive wide interest at the conference. While this problem is challenging and we have not yet solved it, we believe our three major contributions represent important progress:
> We introduce a novel environment and task for embodied reinforcement learning agents to play hide and seek, which we believe will be a valuable resource for the research community to investigate multi-agent visual dynamics.
> Through quantitative experiments, we systematically analyze the emergent representations from the visual hide and seek task. Our results suggest that both the recognition of agency and recognition of self-visibility are automatically learned from our models.
> Furthermore, we investigate why these features emerge from this learning process, and we show that agent weaknesses cause these representations to be learned. Since there is substantial interest in self-supervised learning today, we believe this analysis is timely.
>
> - “I believe the OpenAI environment is also partially observable, where unseen agents’ information is masked out.”
> Thanks for bringing this up, and we double-checked this. Section 4 in Baker et al. [1] says “We utilize decentralized execution and centralized training. At execution time, each agent acts given only its own observations and memory state. At optimization time, we use a centralized omniscient value function for each agent, which has access to the full environment state without any information masked due to visibility” which suggests the environment is fully observable during training. We updated the related work section to clarify this distinction during training.
>
> - “To improve the quality of the paper, is it possible to utilize the features and performance correlation to improve unsupervised visual feature learning?”
> In general, the model will only learn features that are useful for the task at hand (visual hide and seek). We studied this in Section 4.2 and Table 3, which suggests the features are useful for some visual recognition tasks related to the environment.
>
> - “Is it possible to draw a connection to Psychology? Does this Representation vs. Survival Time also happens in real-life, or is it just some empirical things that happen in optimization?”
> We make no claims about the biological plausibility of our approach. We only draw a connection to cognitive science for the overall task of hide and seek, which clearly people are able to play. However, the algorithm and implementation are likely very different. Understanding how biological organisms play hide and seek still remains an open question [2].
>
> - “There is a lot more potential to improve the environments. It will greatly increase the impact of the paper by, for example, considering multi-agent training, self-play, unsupervised training.”
> Thanks for these suggestions. While they are very interesting directions, they are out-of-scope for this paper. We updated our paper in Section 5 to include these as possible directions. This work is our first step towards a better study on these more complicated learning setups, and we hope that our work will shed light on further progress in this problem.
>
> We believe this paper is an important first step towards learning rich representations of multi-agent dynamics. Please kindly let us know if we address your concerns. Thank you!
>
> [1] Bowen Baker, Ingmar Kanitscheider, Todor Markov, Yi Wu, Glenn Powell, Bob McGrew, and Igor Mordatch. Emergent tool use from multi-agent autocurricula, 2019.
> [2] Annika Stefanie Reinhold, Juan Ignacio Sanguinetti-Scheck, Konstantin Hartmann, Michael Brecht, Behavioral and neural correlates of hide-and-seek in rats. Science, 365(6458), pp.1180-1183, 2019

---

> > ### Author Response · Authors · 2019-11-13
> > **Responses and Revisions**
> >
> > Dear Reviewer 2,
> >
> > Thank you again for your constructive reviews! They have helped us improve the quality and clarity of the paper. Based on your reviews, we clarify the related work section and highlighted our contributions in our work. We also updated the paper to include your suggestions on future work and discussed how our work can motivate future research along with these directions.
> >
> > As the discussion period is about to end, please do not hesitate to kindly let us know if there are any additional clarifications or explanations we can offer, as we would love to convince you of the merits of our paper. We appreciate your suggestions and comments! Thank you!

---

### Official Review · AnonReviewer3 · 2019-10-26
**Official Blind Review #3**

**Rating:** 8

**Review:**

# Review ICLR20, Visual Hide and Seek

This review is for the originally uploaded version of this article. Comments from other reviewers and revisions have deliberately not been taken into account. After publishing this review, this reviewer will participate in the forum discussion and help the authors improve the paper.



## Overall

**Summary**

The authors introduce a new RL environment and task, "Visual Hide and Seek", in which they analyze how the agent's learned visual representations are impacted by its speed, auxiliary rewards, and opponent behavior.


**Overall Opinion**

This paper presents a thorough analysis and great visualizations of agent behaviors and representations under different conditions. I wish more papers would put this much effort into analyzing their agents. I'd highly recommend this paper get accepted since I believe the analysis carried out here and the conclusions reached are quite novel and the paper is overall well-written.
However, at the same time, the work of [Baker et al., 2019][1] was published with significantly more fanfare. I hope their work does not overshadow this one since they are only related in the general task concept.

[1]: https://arxiv.org/abs/1909.07528

Some major issues I had with this work:

- In general, please run more random seeds. Just reporting on a single random seed is not enough, as per [Henderson et al., 2018][2].
- There are some sections of the paper where the order of paragraphs is confusing. You start the introduction by stating what you've done and letting the reader wonder "why?". The explanation is only given in the second paragraph. So I'd suggest rotating the second paragraph upwards before the first. Similarly, at the beginning of section 4, you just mention the results - this should either be shorter (1 sentence, as an overview of the work in this section) or moved to the end of that section.
- You're missing a section about future work and flaws/problems of your work at the very end (the latter if which should be in "Discussion"), which is common to include in ICLR publications.

[2]: https://www.aaai.org/ocs/index.php/AAAI/AAAI18/paper/viewFile/16669/16677

Here are some minor...

## Specific comments and questions

### Abstract

all good

### Intro

- Fig.1: What is that white thing that the seeker/hider have on the capsule?

### Rel. Work

all good

### Method

- What's the speed (FPS) of that Unity engine? Why didn't you use Mujuco/(Py)Bullet/Gym-Miniworld?
- You need to add some measurements and units: the arena size doesn't have a unit, the size (diameter) of the hiders/seekers is unclear, turning left/right is unclear (how far left/right, after action-repeat)
- You mention any real-valued position to be valid - so the agents can step into obstacles? And how about through obstacles?
- "Affordance Learning" is usually not used for static environment geometry like obstacles (e.g. [Georgia-Tech course on "Human-Robot Interaction"][3])
- Why 4-layer CNNs? Why not 6 or 8 or a ResNet? Would you think the features would be stronger/weaker in an 8-layer CNN?

[3]: https://www.cc.gatech.edu/~athomaz/classes/CS8803-HRI-Spr08/MayaChandan/Site/Affordance_Learning.html

### Experiments

- 4.1 "... learned this play game" -> "... learned this game".
- 4.2 "mid-level features" -> what's that? The activation of the convolutional kernels after the second layer CNN? What's the dimension? And why did you pick the 2nd layer, not any of the other 3?
- Tab.3: This is averaged over how many frames of rollout?
- "... case can moves a lot faster." -> "... case can move a lot faster".
- Fig.2 is very interesting. Well done.
- Fig.3: the font is not consistent with other figures
- Fig.4: remove the blueish background to increase contrast. Increase the font size of the ticks on the left. Make the legend color boxes slightly bigger. Add more space or a visual divider between the different states - especially on the right side it's hard to make out where one stops and the next begins.
- Fig.4/5: This analysis is lovely and we need more of this in DRL.
- 4.4 in the text, you sometimes write "not S" and sometimes "¬S". Please change the "not s"
- Fig.5: (suggestion) Merge/sum the 2 columns (in both A/B merge the left and right plot into one by summing); subtract the random policy values as you did with Fig.4.
- "We summarize representative cases, and put the full results for all combinations in the Appendix" - no you didn't.
- Fig.6: What is going on in the left third of this diagram? What is this colorful mush? If this is by any chance indicating a change over time, do you maybe want to spread a single, very colorful plot of distance over time into multiple less colorful plots? At least add a legend, please. Also, I'd recommend smoothing (moving average or smoothing spline).
- Also maybe add reward over time plots, as is common in DRL, to show that your policies converged after 8 mil. steps.

### Conclusion

All good, save for the missing future work and critical analysis of your work.

### Appendix

all good

**Experience Assessment:**

I have published one or two papers in this area.

**Review Assessment: Checking Correctness Of Derivations And Theory:**

N/A

**Review Assessment: Checking Correctness Of Experiments:**

I carefully checked the experiments.

**Review Assessment: Thoroughness In Paper Reading:**

I read the paper thoroughly.

---

> ### Author Response · Authors · 2019-11-10
> **Responses to Official Review #3**
>
> Thank you for the helpful suggestions! We appreciate that you recommend our paper and our methods. We have updated our paper to reflect all the changes and please kindly find our responses for each point below:
>
> - “Experiments with more random seed”
> We updated the results to include the standard deviation. Since the relative ranking of models did not change our main conclusions still hold.
>
> - “The order of the first two paragraphs in the introduction and experiment section is confusing”
> Thank you for your suggestions. We updated the order of the paragraphs accordingly. We also shorten the descriptions of the beginning of the experiment section to emphasize just the key messages and the structure of the section.
>
> - We updated our paper in Section 5 to include more future works and discussions.
>
> - “In Figure 1, what is the white thing that the seeker/hider has on the capsule?”
> The white capsule is a visual decoration we have for the agents which we intend to mimic the “mouth,” which is a visual feature to indicate orientation.
>
> - “What is the FPS of that Unity engine? Why didn’t you use Mujoco / (Py)Bullet / Gym-Miniworld?”
> The FPS of our simulation environment is 50. We update our paper to include this. However, this number can be tuned and the limit depends on the computer hardware as well as the complexity of physics simulations for each environment. More information can be found at [1]. Several simulation engines are possible, and we just chose one that is fast and extensible.
>
> - “Add some measurements and units”
> We updated our paper with the measurements and units.
>
> - “You mentioned any real-valued position to be valid - so the agents can step into obstacles?”
> The agents can neither step into the obstacles nor walk through obstacles. We use the built-in collision detection provided by the Unity game engine. We updated our paper to clarify this.
>
> - “The usage of the word “affordance''.
> Thank you for pointing this out. We updated our paper with “recognition”.
>
> - “Number of layers? How about the features in an 8-layer CNN?”
> While several neural network architectures are possible, we follow previous successful implementations with the hyperparameters. Generally, 3 to 5 layers have been shown to be effective for visual observations [2][3][4]. We did not extensively tune hyperparameters.
>
> - We fixed the two typos. Thank you for pointing them out.
>
> - “What is “mid-level” features? What is the dimension?”
> Mid-level features are the output of the last convolutional layer in the policy network. The dimension is 512 as shown in the architecture appendix. We updated our paper to make this explicit. We choose the last convolutional layer right before the visual feature is sent into the two branches of fully-connected layers.
>
>
> - “Table3: this is averaged over how many frames of rollout?”
> We rolled out each policy for at most 50,000 steps. We use a 0.8 train and test split and balance the training and testing set to maintain a random set with 6,000 data points for each task. We updated the paper in Section 4.2 to reflect this.
>
> - Thank you for pointing this out. We changed the font in Figure 3. We polished Figure 4 and update it.
> - Thanks for your nice words for our Figure 4 and 5.
> - We unify all the symbols for "¬S".
>
> - “Merge Figure 5 and subtract the random policy values.”
> Thanks for the suggestion. To make it easier for others in the community to compare to our results, we choose to use absolute values.
>
> - We apologize for the typo. Due to the space of interest, we will release all the 64 transitions plots along with all of our software and code bases. Thanks for catching this and we updated our paper to clarify this.
>
> - For Fig 6, we show the relationship of the distance (y-axis) between the hider and seeker versus time (x-axis). Each color represents one trajectory and we plot 100 trajectories using different colors. We show all of these trajectories together to show the general pattern. The red line is a representative case and we visualize its corresponding trajectory on the left as indicated in the caption. We did not smooth the curve in order to accurately show the changes of the distance and comparisons among different policy variants. We update the caption of Figure 6 and text to clarify these points.
>
> - Thank you for pointing this out. We discussed the potential future work on learning better memory models for our task in the 2nd paragraph of Section 5. We updated our paper with more future works and discussions.
>
> Please kindly let us know if we address your concerns. Thank you!
>
> [1] https://github.com/Unity-Technologies/ml-agents/issues/1425
> [2] Volodymyr Mnih, Koray Kavukcuoglu, David Silver, Andrei A Rusu, Joel Veness, Marc G Belle- mare, Alex Graves, Martin Riedmiller, Andreas K Fidjeland, Georg Ostrovski, et al. Human-level control through deep reinforcement learning. Nature, 518(7540):529, 2015.
> [3] https://github.com/openai/baselines
> [4] https://github.com/ikostrikov/pytorch-a2c-ppo-acktr-gail

---

### Author Response · Authors · 2019-11-10
**Paper Revisions**

We would like to thank all the reviewers for their constructive and helpful suggestions! We appreciate that all the reviewers agree that our proposed task is interesting and can be a valuable resource for the research community. We have updated our paper with a new version that addresses all the suggestions from the reviewers. In summary, the main changes in the paper include:

1) We updated our texts to clarify the facts asked by the reviewers.
2) We reported the standard deviations to include multiple random seeds. Since the ranking of the performances among all the variants does not change, our main conclusions still hold.
3) We added a section in Section 5 to include more future works and discussions on the potential directions of our work.
4) We polished the figures and refined the captions to include more details and clarifications.

---

### Decision · Program_Chairs · 2019-12-19

**Decision:**

Reject

**Comment:**

This paper proposes a technique for training embodied agents to play Visual Hide and Seek where a prey must navigate in a simulated environment in order to avoid capture from a predator. The model is trained to play this game from scratch without any prior knowledge of its visual world, and experiments and visualizations show that a representation of other agents automatically emerges in the learned representation. Results suggest that, although agent weaknesses make the learning problem more challenging, they also cause useful features to emerge in the representation.

While reviewers found the paper explores an interesting direction, concerns were raised that many claims are unjustified. For example, in the discussion phase a reviewer asked how can one infer "hider learns to first turn away from the seeker then run away" from a single transition frequency? Or, the rebuttal mentions "The agent with visibility reward does not get the chance to learn features of self-visibility because of the limited speed hence the model received samples with significantly less variation of its self-visibility, which makes learning to discriminate self-visibility difficult". What is the justification for this? There could be more details in the paper and I'd also like to know if these findings were reached purely by looking at the histograms or by combining visual analysis with the histograms.

I suggest authors address these concerns and provide quantitative results for all of the claims in an improved iteration of this paper.